# The sustainability of recreational sports in Chinese cities based on cognitive entropy

Xuefang Zou[1,2], Jin Wang[3], Xia Zhang[4], Ming Zheng[5], Haitao Chen [6]*

1 College of Graduate, Chengdu Sport University, Chengdu, China, 2 College of Physical Education, Chengdu Normal University, Chengdu, China, 3 College of Education, Zhejiang University, Hangzhou, China, 4 School of Economics and Management, Shanghai University of Sport, Shanghai, China, 5 Zhejiang Academy of Culture and Tourism Development, Tourism College of Zhejiang, Hangzhou, China, 6 College of Physics and Electronic Engineering, Chengdu Normal University, Chengdu, China

* chqcht@sina.com

## Abstract

With progress of science and development of economy, increasing numbers of urban residents take part in recreational sports (RS) due to their more leisure time and increased income. The RS is defined as a kind of physical activities engaged by people during their leisure time for obtaining experience of physical and mental pleasure. The sustainability of the RS plays a critical role in sustainable urban development. As major Chinese cities, such as Hangzhou, Shanghai and Chengdu, are faced with how to maintain sustainable development of their RS systems, which are selected as case cities for empirical analysis. The RS system and its evaluation indicators are constructed by using the conceptual composition method. The RS system is made up of five subsystems including the industrial subsystem, space subsystem, experience subsystem and formal subsystem. The indicator system for assessment of RS system is included 28 indicators. The validity of the RS system questionnaire is tested by employing confirmatory factor analysis and structural equation modeling software. The result indicates good fit of the questionnaire factors. The inquiry was carried out in the three cities where 1080 residents were approached and 1028 questionnaires were collected. The cognitive entropies and weight matrix for the RS system of each sample cities are calculated based on entropy theory. The sustainable development levels of each subsystem of the RS system are obtained by using TOPSIS Method. Then, suggestions of urban planning for three cities have been derived by the results of analysis based on their historical background, regional characteristic and development level.

## 1. Introduction

The recreational sports (RS) is defined as a kind of physical activities engaged by people during their leisure time for obtaining experience of physical and mental

**Data availability statement:** All relevant data are within the paper.

**Funding:** The Ministry of education of Humanities and Social Science project (25YJA890004); On-the-job Doctoral Program for Chengdu Normal University (ZZBS202407); Sichuan Provincial Research Center for Elementary and Middle School Teachers' Ethics, CJSD2401(111/111180127)." The funders provide funding in data collection, analysis, and payment for publication of this paper.

**Competing interests:** The authors have declared that no competing interests exist.

pleasure [1]. From a perspective of social phenomenon, the RS manifests as a social form with the participation of people in sports activities, and gradually evolves into a modern way of life by changing their living habits. Unlike traditional physical sports, participation in recreational sports means "play" rather than "exercise" and thus has a greater potential to be enjoyable [2].

The World Commission on Environment and Development in the report Our Common Future defined the sustainable development as the development that meets the needs the present without compromising the ability of future generations to meet their own needs [3,4]. The sustainability of the RS plays a critical role in sustainable urban development. With progress of science and technology, increasing numbers of Chinese urban people take part in RS due to their more leisure time and increased income. Recently, some scholars discussed the development trend of the RS but their researches confused the concept of the development with the sustainable development [5,6]. Many researches paid attention on one specific aspect of the sustainability assessment of the RS, such as space, experience, form, support or industry, and their results only reveal some limited peculiarities due to their narrow perspective [7–9].

Though sustainability studies are popular topic for researchers in a wide range of fields, it is not easily measurable because it is not directly indicated as a direct consequence of the indicators. Zinatizadeh et. al. applied the Shannon's entropy to construct composite indicators, which were used to measure the urban sustainability [10]. Munier developed a set of sustainability indicators to measure the state of a city, where the entropy method was used for weighting the selected indicators [11]. Lin et al. used the information entropy to analyze the sustainable ability of the urban ecosystem [12].

Recently, some studies used the method which combines the information entropies with the technique for order preference by similarity to ideal solution (TOPSIS) to measure sustainability at different areas [13–17]. For examples, Ding et al. used the TOPSIS-entropy method to evaluate the sustainable development of cities in China [13]. By taking Shandong Province as a case study, Qiao et al. evaluated the urban renewal by using entropy and TOPSIS method [14]. Taking Chengdu as an example, Li et. al. applied the entropy and TOPSIS theory to establish an evaluation system for sustainable agricultural development of Chengdu based on five factors, such as economy, society, environment, education, and population [15]. Wang et. al. constructed an evaluation system to evaluate the agricultural extension service for sustainable agricultural development by using a Hybrid Entropy and TOPSIS Method [16]. In order to evaluate sustainable development of islands, Zhao et. al. developed an evaluation indicator system based on the entropy and TOPSIS models to analysis the effectiveness of the system [17].

On the other hand, most of the studies in the RS have been performed by qualitative analysis methods [5–9], so it's urgent to find out an accurate method to quantitatively assess the sustainability of the urban RS. As a kind of social behaviors in the social system, it would be more consistent with its social property to study the RS from the view of subjective social psychology [18–20]. In the field of leisure science,

researchers pointed out that SD of urban tourism and the sustainable growth of sports affairs are both influenced by their respective entropy [21,22]. These successful researches provide a valuable theoretical perspective in the study of the sustainability of RS.

In this new century, China has promulgated many policies to develop the RS for urban residents. As a developing country, however, the RS system of Chinese cities faces enormous challenges and problems. Therefore, major Chinese cities, such as Hangzhou, Shanghai and Chengdu, are also faced with how to maintain sustainable development of their RS systems, which are selected as case cities for empirical analysis. According to the social system entropy theory, from a "bottom-up" perspective, the development of a social system can be judged by cognitive entropies derived from the perception and satisfaction of social members to the social system [23,24]. The purpose of this research is to evaluate the sustainability of RS in Chinese cities by using concept of the cognitive entropy and TOPSIS Method. Section 2 constructed the RS system and its evaluation indicators. After the inquiry was carried out in Hangzhou, Chengdu, and Shanghai, respectively, the data were collected in Section 3. In Section 4, the RS system of each sample cities was be evaluated of based entropy theory and TOPSIS Method. Then, suggestions for urban planning were be derived by the results of analysis. The main conclusions obtained from this research are concluded in Section 6.

## 2. Construction of the RS system

### 2.1. Construction of RS system

The RS system can be constructed as a social system due to its social and complex nature. During the RS system constructing, the conceptual composition method is employed, which involves graphically representing the structural features of concepts and their interrelationships within the system. By generating lexical flows, categorizing them, and applying multidimensional scaling, structural conceptual mappings of the dimensions of the RS system is constructed.

The concept map is a node-link graph in which every node expresses one concept and each link represents a relationship of two concepts it connects [25]. This approach integrates the strengths of mixed-text analysis and coding methods, and can effectively minimize the reliability challenges come from relying solely on textual explanations or coding [26]. The process of concept mapping involves six key steps:

Step 1: Data collection. With the theme of the RS characteristics, an initial search retrieved 134 relevant research papers. 119 articles were retained based on a review of their titles after removal of 15 duplicate entries. Following an evaluation of the abstracts, 86 papers deemed highly relevant to the topic were selected for further analysis.

Step 2: Vocabulary Streams construction. A vocabulary stream is generated by a sentence where each stream focuses on a noun concept. Sentences that do not describe RS characteristics are excluded from consideration. The construction of vocabulary streams is concerned primarily with conclusions and abstracts in the literatures which reflect RS characteristics. All relevant sentences in them are encoded separately using Nvivo11 analysis software, which result in a total of 245 codes. Then, a high-frequency vocabulary associated with RS characteristics is selected, and finally 28 groups of high-frequency terms are chosen according to each of whom appears more than five times.

Step 3: Vocabulary streams classification. Twenty scholars from the field of physical education were invited to classify the 28 high-frequency terms related to RS characteristics. The scholars were informed of the classification requirements: each high-frequency term should only be classified once and provided appropriate name. Then, the printed materials containing the high-frequency vocabulary were distributed to the scholars, who were asked to categorize the terms.

Step 4: Multidimensional scaling. The vocabulary flow is arranged into rows and columns, creating a $28 \times 28$ two-dimensional matrix (X), in which the matrix elements $X_{ij}$ indicates entries i and j to be placed in the same category by scholars (1) or not (0). The 20 two-dimensional matrices are then superimposed to be a new matrix. The maximum value of 20 and the minimum value of 0 represent the cases which all scholars grouped the entries together or kept them separate, respectively. In this way, each entry receives a two-dimensional coordinate. Based on similarity, a two-dimensional

plot with 28 points are obtained by use of the multidimensional scaling method, in which points representing similar units appear closer together.

Step 5: Cluster analysis and category naming. Based on the results of the above multidimensional scaling, cluster analysis is performed by use of the Ward's method. Previous researches on RS system have used structures ranging from three to thirteen categories [27,28], so this study selects cluster categories within this range. Initially, all items are merged into one category, which results in the outputs ranging from three to thirteen categories. After that, a tree diagram showing the vocabulary flow for each category is constructed.

Subsequently, ten experts were invited to name the 28 entries through a meeting. Based on their review of the materials, the names were determined after two rounds of discussion. According to the clustering analysis order, conceptual terms were discussed and suggested as dimension names in the first round. Once there were no objections, the next dimension naming was proceeded. In order to ensure consistency and clarity of the category, the second round mainly involved revisiting the dimension names until consensus was reached.

Step 6: Construction of the concept map. The centroid coordinates are calculated based on the distribution of all points in the multidimensional matrix. After coordinates (x and y) of each category are determined, the distances between these points and centroid can be calculated, respectively. Categories closer to the centroid are considered more closely related. That is to say, the greater the distance of various categories is in the concept map, the weaker the correlation of them is. Therefore, the map visually represents the spatial distances of every category, offering insight into their relationships. On the hand, each category occupies a different area in the composition, with larger areas indicating that the category encompasses a broader range of related content.

The results of the multidimensional scaling for the 28×28 matrix showed a stress value of 0.08. It is known that a lower stress value indicates higher similarity of the matrix with the map attempted to construct. The stress value of 0.08 in this research is significantly lower than the average stress value of 0.29 typically found in conceptual frameworks and is also below the psychological standard of 0.1 [26], thereby meeting the criteria for its conceptual framework validity. The primary dimensions in the RS structure were named according to the concept closest to the center of mass, combined with the names of adjacent points. This study consists of 5 dimensions and 28 points (expressed by different letters), as shown in Fig 1.

The RS system consists of five distinct sub-structures, each of which is composed of some indicators. Table 1 show the implications of letters in Fig 1.

## 2.2. Test of the questionnaire's reliability

A total of 340 questionnaires were initially distributed at the start of the survey, from which 334 valid ones were retrieved after excluding incomplete responses. Of the valid responses, 172 were from male subjects while 162 were from female subjects. To assess the reliability of the questionnaire, an internal consistency reliability test was conducted using two methods: Cronbach's alpha coefficient (homogeneity reliability) and Guttman Split-half method (split-half reliability) indicated in Table 2. The results show that the questionnaire has good reliability. The overall internal consistency coefficients for the RS perception questionnaire and its five dimensions range from 0.76 to 0.86. Additionally, after the questionnaire was split into two equal halves, the correlation coefficients of them showed that the split-half reliability were found from 0.69 to 0.75. The results suggest that the questionnaire exhibits good internal consistency and reliability, confirming its effectiveness as a measurement tool.

The common method to determine the content validity is that relevant experts are invited to assess representativeness and suitability of the questionnaire items to the intended content scope. In this research, the indicators of the RS system questionnaire were matched with the contents come from relevant literatures and expert interviews, which guided the construction of the questionnaire. Before doing the official survey, a pre-test was conducted with 67 participants to ensure the questionnaire was easy to understand and consistent with the theoretical assumptions of the study. These steps were taken to ensure the content validity of the survey.

 

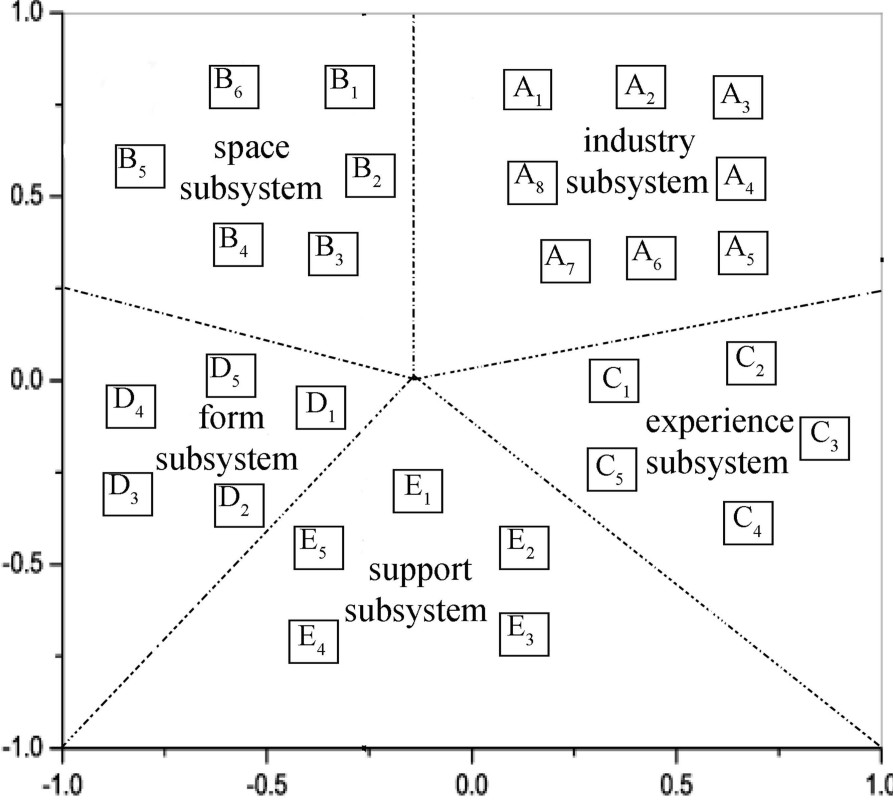

**Fig 1. Conceptual structure of the RS system.**

In order to further test the validity of the RS system questionnaire, we employed confirmatory factor analysis (CFA) by using 16 elements of a second-order dimension based on the preliminary data. The analysis was conducted using AMOS 20.0 structural equation modeling software, which conceptualized and validated the fit of the RS system questionnaire indicators. The overall fit of the questionnaire is included the model fit, determinant fit index, and relative fit index, which are shown in Table 3. The result indicates good fit of the questionnaire factors. In terms of the absolute fit index, the fit index result is 4.20, which meets the requirements of fitting inspection. The root mean square error of approximation (RMSEA) is 0.06, fallen within the acceptable range of 0.05 to 0.08, indicating an acceptable fit of it. The values of goodness-of-fit (GFI) and the adjusted goodness-of-fit (AGFI) are 0.94 and 0.92 respectively, both indicating a very good fit of them [29]. The relative fit index, the normed fit index (NFI), incremental fit index (IFI) and comparative fit index (CFI) are in order as 0.94, 0.95 and 0.95, indicating excellent fit of them because all they are greater than 0.90. The result of the factor analysis aligns well with the theoretical construction factors, confirming the structural validity of the questionnaire. Consequently, the RS perception scale has met the required validity standards. Through this series of analyses, the final RS perception questionnaire was developed, consisting of five secondary dimensions and 28 elements.

## 3. Data collection

### 3.1. Research context

This paper focuses on three major Chinese cities, such as Shanghai, Hangzhou, and Chengdu. Shanghai consistently ranks first in China for its economic vitality and international reputation. As the capital city of Zhejiang Province located in

**Table 1. Indicators of the RS system.**

| First level indexes | | Second-level indexes | |
|---|---|---|---|
| **Name** | **Reference** | **Name** | **Reference** |
| Industrial subsystem | A | Develop sports brand strategy | $A_1$ |
| | | Strengthening the skills training of RS enterprise | $A_2$ |
| | | Service personnel | $A_3$ |
| | | The development of Internet consulting | $A_4$ |
| | | Development industry | $A_5$ |
| | | Establishing the RS industry base and improving the insurance market | $A_6$ |
| | | Stimulate consumption | $A_7$ |
| | | Scientific construction of facilities and management | $A_8$ |
| Space subsystem | B | Community sports space | $B_1$ |
| | | Road side space | $B_2$ |
| | | Grass space | $B_3$ |
| | | City parks | $B_4$ |
| | | Residences surrounding parks | $B_5$ |
| | | Sports parks | $B_6$ |
| Experience subsystem | C | Funny of sports participation | $C_1$ |
| | | Enhance one's inner self | $C_2$ |
| | | Realize the value of life | $C_3$ |
| | | Positive effect on psychological well-being | $C_4$ |
| Form subsystem | D | Extreme sports | $D_1$ |
| | | Video games | $D_2$ |
| | | Chess and card game | $D_3$ |
| | | Sports game | $D_4$ |
| | | Spectating sports | $D_5$ |
| Support subsystem | E | Integrate RS into development plan of the region | $E_1$ |
| | | Investment to community RS events | $E_2$ |
| | | The government support development of education, services and funds of RS | $E_3$ |
| | | Introduce social capital into the RS | $E_4$ |
| | | Supervising private RS organizations | $E_5$ |

**Table 2. Reliability of factors related to RS system perception.**

| factor | Industry subsystem | Form subsystem | Experience subsystem | Space subsystem | Supporting subsystem | Overall questionnaire |
|---|---|---|---|---|---|---|
| α coefficient | 0.75 | 0.78 | 0.83 | 0.78 | 0.89 | 0.87 |
| split half reliability | 0.74 | 0.69 | 0.74 | 0.74 | 0.68 | 0.82 |

**Table 3. Summary of the fitting and validation index values of the questionnaire.**

| model | Decide on the fit index | | | | | | Relative fit index | | |
|---|---|---|---|---|---|---|---|---|---|
| | $x^2$ | $df$ | $x^2/df$ | RMSEA | GFI | AGFI | NFI | IFI | CFI |
| physical culture arder system | 394.97 | 94 | 4.20 | 0.06 | 0.94 | 0.92 | 0.94 | 0.95 | 0.95 |

eastern China, Hangzhou is second-tier cities due to its economic level. Although it is located at western China, Chengdu is the fourth populous city in China and has successfully hosted the world university student games and the World Games.

### 3.2. Data collection

The name of the institutional review board or ethics committee that approved the study is Institutional Review Board of Psychology and Behavior of Chengdu Normal University. Approval number of this research is CDNU2023032. The form of consent obtained: oral. Before questionnaires for this study, the purpose of the investigation was explained to the respondents. The inquiry was carried out from August to October 2023 in Hangzhou, Chengdu, and Shanghai, respectively. Respondents were randomly selected on weekends, during which there were more people appearing in the park. A total of 1028 questionnaires were collected and the response rate was 89.7%. 542 (51.9%) of them were male and 486 (48.1%) were female. Respondents ranged from 18 to 80 years old with a mean age of 35.6 years.

### 3.3. Measures

For measuring the people' satisfaction with the RS cognition of their social system, the questionnaire designed includes the cognition and satisfaction of the RS system (28 items).

## 4. Evaluation of the RS system

### 4.1. Cognitive entropy

Entropy is a concept of physics, related with the degree of disorder of a state, which is equal to the ratio of the heat absorbed by the system to the temperature and is expressed as [30]

$$dS = \frac{dQ}{T},$$
(1)

In the twentieth century, Shannon defines entropy of information $H(x)$ as [31]

$$H(x) = -p(x) \ln p(x),$$
(2)

where $p(x)$ is the probability of the event $(x)$. It is seen from Equation (2) that $H$ is always non-negative, which is used to measure randomness of a system.

Recently, scholars proposed that the cognitive entropy, may be used to explain the process of energy transformation in human social activities [32, 33]. According to the social system entropy theory, the development status of a social system can be evaluated by cognitive entropies deriving from the perception and satisfaction of social members to the social system [34,35]. Cognitive entropy refers to the degree of satisfaction of members of a society with the development level of the social system based on their understanding of it [34,36]. In this research, as a type of emotional responses, cognitive entropy is the social members' cognition-satisfaction assessment with the RS system, which is expressed as

$$h_{mn} = \frac{S_{mn}}{C_{mn}},$$
(3)

where $C_{mn}$ and $S_{mn}$ are the people's cognition and satisfaction with each indicator of the RS system, respectively.

For each subsystem of the Rs system, entropy is calculated by using the following formula, respectively [25]

$$H_m = -\frac{1}{\ln N} \sum_{n=1}^{N} h_{mn} \ln h_{mn},$$
(4)

where $H_m$ expresses the social cognitive entropy of a subsystem and $N$ is the number of its categories, as well as $h_{mn}$ is the probability of occurrence of each category in which can be calculated by Equation (3).

In order to eliminate the influence of different subsystem on the total entropy of the RS system, the weight $w_m$ is the weight of a subsystem evaluation index, the calculation method is as follow [14]

$$w_m = \frac{1 - H_m}{\sum_{m=1}^{M} (1 - H_m)},$$

(5)

where $M$ is the number of subsystems of the RS system.

By using Formulas (3) – (5), Table 4 shows the calculation results of cognitive entropies and weight matrix for the RS system.

**Table 4. Cognitive entropy and weight matrix for the RS system.**

| Indicator | | $h_{mn}$ | | | $H_m$ | | | $w_m$ | | |
|---|---|---|---|---|---|---|---|---|---|---|
| | | I | Π | ш | I | Π | ш | I | Π | ш |
| A | $A_1$ | 0.15 | 0.05 | 0.20 | 0.78 | 0.82 | 0.84 | 0.23 | 0.13 | 0.18 |
| | $A_2$ | 0.21 | 0.07 | 0.08 | | | | | | |
| | $A_3$ | 0.20 | 0.05 | 0.21 | | | | | | |
| | $A_4$ | 0.02 | 0.07 | 0.14 | | | | | | |
| | $A_5$ | 0.05 | 0.10 | 0.05 | | | | | | |
| | $A_6$ | 0.02 | 0.14 | 0.07 | | | | | | |
| | $A_7$ | 0.15 | 0.13 | 0.03 | | | | | | |
| | $A_8$ | 0.12 | 0.11 | 0.06 | | | | | | |
| B | $B_1$ | 0.14 | 0.20 | 0.24 | 0.68 | 0.65 | 0.97 | 0.33 | 0.24 | 0.03 |
| | $B_2$ | 0.07 | 0.20 | 0.22 | | | | | | |
| | $B_3$ | 0.04 | 0.10 | 0.20 | | | | | | |
| | $B_4$ | 0.10 | 0.13 | 0.21 | | | | | | |
| | $B_5$ | 0.05 | 0.05 | 0.10 | | | | | | |
| | $B_6$ | 0.11 | 0.03 | 0.09 | | | | | | |
| C | $C_1$ | 0.04 | 0.05 | 0.25 | 0.64 | 0.43 | 0.94 | 0.38 | 0.40 | 0.07 |
| | $C_2$ | 0.10 | 0.10 | 0.45 | | | | | | |
| | $C_3$ | 0.10 | 0.10 | 0.40 | | | | | | |
| | $C_4$ | 0.18 | 0.26 | 0.12 | | | | | | |
| D | $D_1$ | 0.27 | 0.17 | 0.11 | 0.96 | 0.96 | 0.71 | 0.04 | 0.03 | 0.32 |
| | $D_2$ | 0.21 | 0.20 | 0.07 | | | | | | |
| | $D_3$ | 0.26 | 0.14 | 0.06 | | | | | | |
| | $D_4$ | 0.21 | 0.24 | 0.11 | | | | | | |
| | $D_5$ | 0.10 | 0.21 | 0.20 | | | | | | |
| E | $E_1$ | 0.27 | 0.27 | 0.05 | 0.98 | 0.72 | 0.63 | 0.02 | 0.20 | 0.40 |
| | $E_2$ | 0.24 | 0.05 | 0.18 | | | | | | |
| | $E_3$ | 0.20 | 0.17 | 0.11 | | | | | | |
| | $E_4$ | 0.26 | 0.22 | 0.11 | | | | | | |
| | $E_5$ | 0.10 | 0.01 | 0.03 | | | | | | |

Note: Greek letters I, Π and ш express Shanghai, Hangzhou and Chengdu, respectively.

## 4.2. Sustainable development evaluation

The approach of technique for order preference by similarity to an ideal solution proposed (TOPSIS) by Hwang and Yoon [37] can make us formulate a positive ideal solution and a negative ideal solution. With this approach, the distance between the scheme and the ideal solutions can be calculated, which can be used as a criterion to compare different schemes. The main steps are as follows [14–17].

Step 1, as normalized value of the cognitive entropy $h_{mn}$, $H_{mn}$ is expressed as

$$H_{mn} = \frac{h_{mn}}{\sum\limits_{n=1}^{N} h_{mn}}.$$

(6)

Step 2, normalized weight of each cognitive entropy can be calculated by

$$W_{mn} = w_m H_{mn}.$$

(7)

Step 3, obtain the positive and negative ideal solutions $W_+$ and $W_-$ by using the following formulas

$$W_m^+ = \left\{ Max W_{mn} \, \middle| \, m = 1, 2, \ldots M; n = 1, 2, \ldots N \right\}.$$

(8)

$$W_m^- = \left\{ Min W_{mn} \, \middle| \, m = 1, 2, \ldots M; n = 1, 2, \ldots N \right\}.$$

(9)

Step 4, with using the following formulas, the distances between the solution scheme and the positive and negative ideal solutions can be calculated

$$S_m^+ = \sqrt{\sum_{n=1}^{N} \left( W_{mn} - W_m^+ \right)^2}.$$

(10)

$$S_m^- = \sqrt{\sum_{n=1}^{N} \left( W_{mn} - W_m^- \right)^2}.$$

(11)

By using Formulas (6) – (11) and results of Table 4, Table 5 shows the calculation results of the distances between the solution scheme and the ideal solutions for the RS system.

Step 5, the sustainable development level of each subsystem for the RS system can be acquired by use of the following formula

$$S_m = \frac{S_m^-}{S_m^+ + S_m^-}.$$

(12)

If $S_m$ is closer to 1, the evaluated solution scheme is closer to the ideal solution.

By substituting the calculation results of the distances between the solution scheme and the ideal solutions each subsystem of the RS system in Table 5 into Equation (12), the sustainable development levels of each subsystem of the RS system can be obtained for three cities and shown in Fig 2.

**Table 5. The distances between the solution scheme and the ideal solutions for the RS system.**

| Indicator | | $H_{mn}$ | | | $W_{mn}$ | | | $S_m^+$ | | | $S_m^-$ | | |
|---|---|---|---|---|---|---|---|---|---|---|---|---|---|
| | | I | П | ш | I | П | ш | I | П | ш | I | П | ш |
| A | $A_1$ | 0.16 | 0.07 | 0.23 | 0.04 | 0.01 | 0.04 | 0.09 | 0 | 0.09 | 0.10 | 0.02 | 0.06 |
| | $A_2$ | 0.23 | 0.10 | 0.10 | 0.05 | 0.01 | 0.02 | | | | | | |
| | $A_3$ | 0.22 | 0.07 | 0.25 | 0.05 | 0.01 | 0.05 | | | | | | |
| | $A_4$ | 0.02 | 0.10 | 0.17 | 0.00 | 0.01 | 0.03 | | | | | | |
| | $A_5$ | 0.05 | 0.14 | 0.06 | 0.01 | 0.02 | 0.01 | | | | | | |
| | $A_6$ | 0.02 | 0.19 | 0.08 | 0.00 | 0.02 | 0.01 | | | | | | |
| | $A_7$ | 0.16 | 0.18 | 0.04 | 0.04 | 0.02 | 0.01 | | | | | | |
| | $A_8$ | 0.13 | 0.15 | 0.07 | 0.03 | 0.02 | 0.01 | | | | | | |
| B | $B_1$ | 0.27 | 0.28 | 0.23 | 0.09 | 0.07 | 0.01 | 0.10 | 0.09 | 0.01 | 0.09 | 0.09 | 0.02 |
| | $B_2$ | 0.14 | 0.28 | 0.21 | 0.05 | 0.07 | 0.01 | | | | | | |
| | $B_3$ | 0.08 | 0.14 | 0.19 | 0.03 | 0.03 | 0.01 | | | | | | |
| | $B_4$ | 0.20 | 0.18 | 0.20 | 0.07 | 0.04 | 0.01 | | | | | | |
| | $B_5$ | 0.10 | 0.07 | 0.09 | 0.03 | 0.02 | 0.00 | | | | | | |
| | $B_6$ | 0.22 | 0.04 | 0.08 | 0.07 | 0.01 | 0.00 | | | | | | |
| C | $C_1$ | 0.10 | 0.10 | 0.20 | 0.04 | 0.04 | 0.01 | 0.14 | 0.28 | 0.04 | 0.14 | 0.22 | 0.04 |
| | $C_2$ | 0.24 | 0.20 | 0.37 | 0.09 | 0.08 | 0.03 | | | | | | |
| | $C_3$ | 0.24 | 0.02 | 0.33 | 0.09 | 0.01 | 0.02 | | | | | | |
| | $C_4$ | 0.43 | 0.51 | 0.55 | 0.16 | 0.20 | 0.04 | | | | | | |
| D | $D_1$ | 0.26 | 0.18 | 0.20 | 0.01 | 0.01 | 0.06 | 0.01 | 0.01 | 0.14 | 0.02 | 0.02 | 0.08 |
| | $D_2$ | 0.20 | 0.20 | 0.13 | 0.00 | 0.01 | 0.04 | | | | | | |
| | $D_3$ | 0.25 | 0.15 | 0.11 | 0.01 | 0.00 | 0.04 | | | | | | |
| | $D_4$ | 0.20 | 0.25 | 0.20 | 0.01 | 0.00 | 0.06 | | | | | | |
| | $D_5$ | 0.10 | 0.22 | 0.36 | 0.00 | 0.01 | 0.12 | | | | | | |
| E | $E_1$ | 0.25 | 0.38 | 0.10 | 0.01 | 0.08 | 0.04 | 0.02 | 0.24 | 0.18 | 0.01 | 0.14 | 0.15 |
| | $E_2$ | 0.22 | 0.35 | 0.38 | 0.00 | 0.07 | 0.15 | | | | | | |
| | $E_3$ | 0.19 | 0.24 | 0.23 | 0.00 | 0.05 | 0.09 | | | | | | |
| | $E_4$ | 0.24 | 0.31 | 0.23 | 0.00 | 0.06 | 0.09 | | | | | | |
| | $E_5$ | 0.09 | 0.01 | 0.06 | 0.00 | 0.00 | 0.02 | | | | | | |

Note: Greek letters I, П and ш express Shanghai, Hangzhou and Chengdu, respectively.

From the perspective of industry subsystem, it is seen from Fig 2 that Hangzhou is at the highest level of the sustainable development while Chengdu is at the lowest level among three cities. From the perspective of space subsystem, Chengdu is at the highest level of the sustainable development level while Shanghai is at the lowest level among three cities. Conversely, Chengdu is at the highest level but Hangzhou is at the lowest level in terms of experience subsystem among three cities. Besides, Shanghai is at the highest level but Chengdu is at the lowest level in terms of form subsystem among three cities. However, being quite the other way, Chengdu is at the highest level but Shanghai is at the lowest level in term of support subsystem.

## 5. Suggestions for urban planning

In order to enhance the sustainability of urban sports and leisure activities, it is necessary for local governments to adopt corresponding measures, because each city has its historical background, regional characteristic and development level. Thus, based on the results of the investigation and calculation above, some urban planning suggestions have been proposed for the three cities.

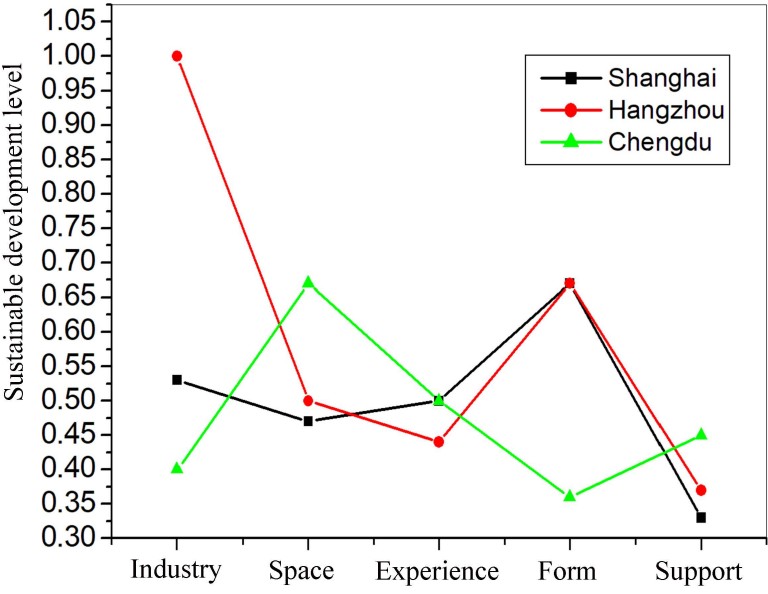

**Fig 2. The sustainable development level of each subsystem of the RS system.**

### 5.1. Shanghai

It is seen from Fig 2 that the formal subsystem is at the highest level of the sustainable development in five subsystems, but the support and space subsystems are at the two lowest levels in them. Only compensating the weaknesses of its subsystems, the RS system of Shanghai may be achieved sustainable development. Firstly, based on indicators for the RS system in Table 2 and the normalized value of their cognitive entropies $H_{mn}$ and weights $W_{mn}$ in Table 5, the reason why its support subsystem is backward lies in its insufficient investment to community RS events, The government support development of education, services and funds of RS, and supervising private RS organizations. Therefore, in urban plannings, it is important for its relevant administrative departments to take steps to advance the development of the support subsystem in these areas. In other words, Shanghai should introduce a series of reasonable policies to promote the development of its RS.

On the other hand, it is seen from Tables 2 and 5 that the reason for the backwardness of the space subsystem in Shanghai lies in its insufficient road side space, city parks and sports parks. Therefore, in urban plannings, it is important to develop public sports space for people to kill time. Though Shanghai is a mega city where every inch of land is extremely valuable, community gardens and pocket sports parks should be constructed in residential area or old factory buildings. Green ways and RS facilities should be built along the rivers or bund of Shanghai, such as Yangtze river, Huangpu river and Suzhou river. Besides, under-bridge and rooftop should be opened up new horizons of three-dimensional movement.

### 5.2. Hangzhou

In Hangzhou, the industry subsystem is at the highest level in five subsystems, but the support subsystem is at the lowest level of the all subsystems. Being similar to the case of Shanghai, it can be seen from Tables 2 and 5 that the reasons of its poor support subsystem are that its insufficient investment to community RS events, the government support development of education, services and funds of RS, and supervising private RS organizations. Thus, in urban plannings, it is important for its relevant administrative departments to introduce a series of reasonable policies to stimulate the development of its RS.

### 5.3. Chengdu

It is seen from Fig 2 that the space subsystem is at the highest level of the sustainable development in five subsystems, but the industrial and form subsystem are at the two lowest levels in them. It is seen from Tables 2 and 5 that the lowest scores of the subsystem's indicators are the skills training of RS enterprise, development of industry, insurance market, stimulate consumption, scientific construction of facilities and management. According to these reasons, the government of Chengdu should adopt measures to strengthen the skills training of RS enterprise, develop insurance market, perfect recreational facilities and improve ability of service in urban plannings. Besides, as a west city, average income of residents is low. Thus, the government should take steps to increase the resident income and stimulate consumption of recreational activity.

Secondly, it is seen from Tables 2 and 5 that the reason for the backwardness of the form subsystem in Chengdu lies in its less forms of sports games. Therefore, in urban plannings, it is important to encourage business to develop various forms of recreational sports for people to kill time. For examples, to the elderly people, the government should encourage RS enterprises to design and install gym equipment with voice-activated features or wearable devices which can real-time monitor physical indicators. For children and teenagers, on the other hand, offer them adventure sport games or facilities with query visualization of growth data. These measures can stimulate people's enthusiasm for RS participation.

## 6. Conclusion

The recreational sports (RS) is defined as a kind of physical activities engaged by people during their leisure time for obtaining experience of physical and mental pleasure. With progress of science and technology, more and more Chinese urban people take part in RS due to their more leisure time and increased income. The sustainability of the RS plays a critical role in sustainable urban development. As a developing country, the RS system of Chinese cities faces enormous challenges and problems. Major Chinese cities, such as Hangzhou, Shanghai and Chengdu, are also faced with how to maintain sustainable development of their RS systems, which are selected as case cities for empirical analysis. According to the social system entropy theory, from a "bottom-up" perspective, the development of a social system can be judged by cognitive entropies derived from the perception and satisfaction of social members to the social system.

The RS system and its evaluation indicators have been constructed by using the conceptual composition method. Meanwhile the validity of the RS system questionnaire has been tested by employing confirmatory factor analysis and structural equation modeling software. The result indicates good fit of the questionnaire factors. After the inquiry was carried out in Hangzhou, Chengdu, and Shanghai, respectively, the data have been collected. After that, the cognitive entropies and weight matrix for the RS system of each sample cities have been calculated based on entropy theory. Then, the sustainable development levels of each subsystem of the RS system have been obtained by using TOPSIS Method. In order to enhance the sustainability of urban sports and leisure activities, suggestions of urban planning for three cities have been derived by the results of analysis and their historical background, regional characteristic and development level.

## 7. Limitations and future research

This study adopts the cross-sectional research method, which is lack of research on the longitudinal trajectory of the cities selected as samples. Due to the results obtained from a one-time survey, there is a lack of follow-up investigation on the development status of the samples. Therefore, the result of this research has certain limitations in implementation and application. Secondly, the categories and scale of the survey subjects are limited, which lead to the results might be inaccurate. This survey only covered urban residents and the sample size of each city is only a few hundred residents. Besides, the research did not survey the perceptions of government officials and urban managers, which would lead to some different results might if such groups were included in the survey. Besides, due to the lack of research resources and research time, only three cities were selected as the survey subjects in this research. However, the randomness and representativeness of the sample may be affected by geographical factors of the selected cities.

Thus, future research should enlarge the sample size by selecting more representative cities and the survey population (including government officials and urban managers). If these points are taken into the consideration, the future study conclusions should be more in line with reality and practical application value.

## Supporting information

**S1 Appendix. Informed Consent Form.**
(DOCX)

## Author contributions

**Conceptualization:** Xuefang Zou, Jin Wang, Haitao Chen.

**Data curation:** Xuefang Zou, Xia Zhang.

**Formal analysis:** Xuefang Zou.

**Funding acquisition:** Xuefang Zou, Jin Wang, Haitao Chen.

**Investigation:** Xuefang Zou, Xia Zhang, Ming Zheng.

**Methodology:** Xuefang Zou, Jin Wang, Haitao Chen.

**Project administration:** Jin Wang.

**Software:** Haitao Chen.

**Supervision:** Jin Wang, Haitao Chen.

**Writing – original draft:** Xuefang Zou.

**Writing – review & editing:** Jin Wang, Haitao Chen.

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
