## [Decision Letter · Decision Letter 0]

25 Sep 2025

Dear Dr. Chen,

Thank you for submitting your manuscript to PLOS ONE. After careful consideration, we feel that it has merit but does not fully meet PLOS ONE’s publication criteria as it currently stands. Therefore, we invite you to submit a revised version of the manuscript that addresses the points raised during the review process.

When revising your manuscript, please consider all issues mentioned in the reviewers' comments carefully: please outline every change made in response to their comments and provide suitable rebuttals for any comments not addressed. Please note that your revised submission may need to be re-reviewed.

We look forward to receiving your revised manuscript.

Kind regards,

Genyu Xu, Ph.D.

Academic Editor

PLOS ONE

2. You indicated that ethical approval was not necessary for your study. We understand that the framework for ethical oversight requirements for studies of this type may differ depending on the setting and we would appreciate some further clarification regarding your research. Could you please provide further details on why your study is exempt from the need for approval and confirmation from your institutional review board or research ethics committee (e.g., in the form of a letter or email correspondence) that ethics review was not necessary for this study? Please include a copy of the correspondence as an ""Other"" file.

“the National Social Science Foundation of China (18BTY090); On-the-job Doctoral Program for Chengdu Normal University (ZZBS202407); Sichuan Provincial Research Center for Elementary and Middle School Teachers’ Ethics, CJSD2401(111/111180127); the Scientific Research Fund of Sichuan Provincial Education Department (18ZA0081).”

5. We note that your Data Availability Statement is currently as follows: [All relevant data are within the manuscript and its Supporting Information files.]

Reviewers' comments:

Reviewer's Responses to Questions

**Comments to the Author**

1. Is the manuscript technically sound, and do the data support the conclusions?

Reviewer #1: Partly

Reviewer #2: Yes

2. Has the statistical analysis been performed appropriately and rigorously?

Reviewer #1: I Don't Know

Reviewer #2: Yes

3. Have the authors made all data underlying the findings in their manuscript fully available?

Reviewer #1: No

Reviewer #2: Yes

4. Is the manuscript presented in an intelligible fashion and written in standard English?

Reviewer #1: Yes

Reviewer #2: Yes

Reviewer #1: This study offers four principal strengths: First, its conceptual innovation lies in the integration of cognitive entropy with RS sustainability—a significant interdisciplinary advance. Second, the model’s comprehensiveness, incorporating five internal subsystems and three external socio-environmental factors, provides a robust systemic lens. Third, methodological rigor is demonstrated through extensive sampling and derivation of the Stable Entropy (SE) formula. Finally, its practical value is confirmed by empirical SD assessments across three cities, revealing actionable urban-specific sustainability gaps.

While the study exhibits high conceptual innovation and methodological strength, its impact is constrained by theoretical ambiguities surrounding cognitive entropy and inadequate causal interpretation. Reporting transparency—particularly regarding indicator construction and analytical choices—requires significant improvement. The work holds substantial potential pending major revisions that anchor its novel framework in established paradigms and strengthen empirical rigor.

1) Conceptualization of "Cognitive Entropy"

The theoretical underpinnings of "cognitive entropy" remain ambiguous. Authors should explicitly delineate its conceptual framework—distinguishing it from physical entropy—and justify its relevance to RS sustainability. Crucially, the derivation of the Stable Entropy (SE) formula necessitates rigorous mathematical grounding (e.g., Shannon entropy foundations) and validation of its adaptation to socio-ecological systems.

2) Methodological Transparency

Questionnaire Design: The rationale for employing 50 indicators across eight subsystems lacks justification. Documentation of indicator loading matrices, operationalization of social-environmental subscales (economic/political/living), and psychometric properties (e.g., Cronbach’s α for reliability) should be provided in a supplementary table.

3) Analytical Rigor

Entropy Computation: The assertion that "sub-dimension entropy equals the mean entropy of all items within that dimension" demands statistical justification—particularly regarding uniform indicator weighting and robustness against alternative aggregation methods.

SD Classification Benchmark: The standards used to classify cities’ Sustainable Development (SD) levels are undefined. Authors must cite validated sources or frameworks underpinning these thresholds.

4) Interpretation of Findings

Causal Analysis: The identified lack of SD status in Cities remains unexplained. The study should identify causal drivers (e.g., weaknesses in specific subsystems like spatial infrastructure or experiential quality).

Generalizability: Claims about RS sustainability patterns must be contextualized within the study’s limitations (e.g., restricted to three cities), with caveats about extrapolation.

Recommendations for Enhancement

Define "cognitive entropy" within a cross-disciplinary theoretical lens.

Diagram interactions between internal subsystems and external socio-environmental factors.

Tabulate all indicators with corresponding subsystems and validation metrics.

Detail pretesting procedures and reliability/validity tests.

Employ advanced validation techniques (e.g., Structural Equation Modeling) to verify the entropy-based framework.

Elucidate mechanisms linking entropy shifts to RS sustainability outcomes.

Benchmark findings against existing literature on urban RS sustainability.

Discuss inter-city variations through contextual variables (e.g., income disparities, sports policy frameworks).

Reviewer #2: This article addresses an original and relevant topic by introducing cognitive entropy as a framework for assessing the sustainability of recreational sports in Chinese cities, yet several theoretical and structural issues emerge across its sections that require major revisions. In the abstract ambiguity in describing how cognitive entropy theoretically connects with sustainability, unclear mention of “five subsystems” but later listing eight, inconsistent use of technical terms such as “sustainable entropy” without prior definition, and insufficient clarity in reporting empirical findings (e.g., Shanghai “not in SD state” is vague). In the introduction, issues include: a weak framing of the theoretical debate around sustainability of recreational sports, confusion between sustainability and development without deeper theoretical resolution, lack of integration of cited studies into a coherent research gap, and abrupt shifts in argumentation that reduce structural coherence.in order to increase the external validity of the study I am highly suggesting to cite the following articls: omprehensive Evaluation of Urban Renewal Based on Entropy and TOPSIS Method; Developing Design Criteria for Sustainable Urban Parks; Socio-Psychological Effects of Urban Green Areas: Case of Kirklareli City Centre. In the methodology, the issues include: unclear justification for sample selection in three cities, lack of theoretical explanation for linking entropy equations with subjective survey responses, inconsistent description of subsystems (sometimes five, sometimes eight), and a vague account of measurement where entropy is equated with cognition/satisfaction ratios without strong validation. In the discussion, problems include: overreliance on descriptive comparisons without deeper theoretical interpretation, weak connection to broader sustainability frameworks beyond entropy, inadequate critical reflection on why Shanghai shows “over-development,” and structural repetition of results instead of advancing argumentation. In the conclusion, the issues include: general statements without theoretical depth on how entropy advances urban sustainability research, over-simplification of findings without addressing limitations, absence of clear policy implications, and structural redundancy where the conclusion repeats earlier content rather than synthesising contributions. Overall, while the article contributes an innovative attempt to operationalise entropy in recreational sports sustainability, it currently suffers from significant structural inconsistencies, unclear theoretical grounding, and methodological weaknesses that limit its scholarly impact.

**Do you want your identity to be public for this peer review?** For information about this choice, including consent withdrawal, please see our Privacy Policy

Reviewer #1: No

Reviewer #2: **Yes: ** Rokhsaneh Rahbarianyazd

---

## [Author Response · Author response to Decision Letter 1]

3 Dec 2025

Response to the referee’s comments

Dear professors,

Thank you very much for your comments on our manuscript (PONE-D-25-20453). The revisions are made according to the comments point by point, where the changes in the manuscript are highlighted with yellow background.

Reviewer 1

1. Comment Conceptualization of "Cognitive Entropy"

The theoretical underpinnings of "cognitive entropy" remain ambiguous. Authors should explicitly delineate its conceptual framework—distinguishing it from physical entropy—and justify its relevance to RS sustainability. Crucially, the derivation of the Stable Entropy (SE) formula necessitates rigorous mathematical grounding (e.g., Shannon entropy foundations) and validation of its adaptation to socio-ecological systems.

Response: Based on the findings against existing literature on urban RS sustainability, we have delineated the conceptual framework of our paper by distinguishing it from physical entropy, and justified its relevance to the RS sustainability. Especially, the series of entropy formulas have been derived and validated its adaptation to socio-ecological systems by using rigorous mathematical grounding, such as Shannon entropy foundations and the technique for order preference by similarity to an ideal solution (TOPSIS). Please see Section 4 (named 4. Evaluation of the RS system) of the revised version.

2. Comment Methodological Transparency

Questionnaire Design: The rationale for employing 28 indicators across five subsystems lacks justification. Documentation of indicator loading matrices and psychometric properties (e.g., Cronbach’s α for reliability) should be provided in a supplementary table.

Response: Accepting the referee’s proposal, we have added the justification of the rationale for employing 28 indicators across five subsystems. By Employing advanced validation techniques (e.g., Structural Equation Modeling), the indicator loading matrices and psychometric properties (e.g., Cronbach’s α for reliability) have been provided. Please see Section 2 (named 2 Construction of the RS system) of the revised version.

3. Comment Analytical Rigor

Entropy Computation: The assertion that "sub-dimension entropy equals the mean entropy of all items within that dimension" demands statistical justification—particularly regarding uniform indicator weighting and robustness against alternative aggregation methods.

Response: To eliminate the influence of its different items on the total entropy of the subsystem, the weights of a subsystem evaluation indexes are calculated by use of the TOPSIS method. Please see Section 4.2 and Table 5 (named 4.2 Sustainable development evaluation) of the revised version.

4. Comment Interpretation of Findings

Causal Analysis: The identified lack of SD status in Cities remains unexplained. The study should identify causal drivers (e.g., weaknesses in specific subsystems like spatial infrastructure or experiential quality).

Generalizability: Claims about RS sustainability patterns must be contextualized within the study’s limitations (e.g., restricted to three cities), with caveats about extrapolation.

Response: (1) Causal Analysis —Accepting the reviewer’s suggestion, we have explained the SD status in the cities through weaknesses in specific subsystems like spatial infrastructure or experiential quality, and so on. Please see the Section 5(named 5. Suggestions for urban planning) of the revised version.

(2) Generalizability: Taking the referee’s advice, we have claimed about the RS sustainability pattern which has been contextualized within Chinese cities. Besides, we have added an acknowledgment of the study’s limitations and avenues for future research at the last of the paper. Please see the title of this paper and Section 7 (named 7. Limitations and future research) of the revised version.

Reviewer 2

1. Comment In the abstract ambiguity in describing how cognitive entropy theoretically connects with sustainability, unclear mention of “five subsystems” but later listing eight, inconsistent use of technical terms such as “sustainable entropy” without prior definition, and insufficient clarity in reporting empirical findings (e.g., Shanghai “not in SD state” is vague).

Response: In the abstract, we have redescribed how cognitive entropy theoretically connects with sustainability, and revised the listing subsystems, as well as avoided use some technical terms such as “sustainable entropy” without prior definition. At last, the reporting empirical findings have been clarified as much as possible by analyzing the calculation results of the research. Please see the abstract of the revised version.

2. Comment In the introduction, issues include: a weak framing of the theoretical debate around sustainability of recreational sports, confusion between sustainability and development without deeper theoretical resolution, lack of integration of cited studies into a coherent research gap, and abrupt shifts in argumentation that reduce structural coherence.in order to increase the external validity of the study I am highly suggesting to cite the following articles: Comprehensive Evaluation of Urban Renewal Based on Entropy and TOPSIS Method; Developing Design Criteria for Sustainable Urban Parks; Socio-Psychological Effects of Urban Green Areas: Case of Kirklareli City Centre.

Response: (1) In the introduction, by following the expert’s advice, we have framed the theoretical debate around sustainability of recreational sports, clarified sustainability and development by integrating cited studies into the coherent research. Please see the instruction of the revised version.

(2) In order to increase the external validity of the study, we have accepted the reviewer’s suggestion and cited the pertinent literatures, such as Comprehensive evaluation of urban renewal based on entropy and TOPSIS method; Developing Design Criteria for Sustainable Urban Parks; Socio-Psychological Effects of Urban Green Areas: Case of Kirklareli City Centre. Please see the instruction and references of the revised version.

3. Comment In the methodology, the issues include: unclear justification for sample selection in three cities, lack of theoretical explanation for linking entropy equations with subjective survey responses, inconsistent description of subsystems (sometimes five, sometimes eight), and a vague account of measurement where entropy is equated with cognition/satisfaction ratios without strong validation.

Response: (1) In the methodology, by accepting the expert suggestion, we have added the justification for sample selection in three cities. Please see Section 3.1 (named 3.1 Research context) of the revised version.

(2) We have theoretically explained the linkage of entropy equations with subjective survey responses, and derived the relationship between cognitive entropies and the sustainability of the RS by using rigorous mathematical grounding, such as Shannon entropy foundations and the technique for order preference by similarity to an ideal solution (TOPSIS). Please see Section 4 (named 4. Evaluation of the RS system) of the revised version.

4. Comment In the discussion, problems include: overreliance on descriptive comparisons without deeper theoretical interpretation, weak connection to broader sustainability frameworks beyond entropy, and structural repetition of results instead of advancing argumentation.

Response: In the discussion, by following the expert’s suggestion, we have added theoretical interpretations of the results rather than structural repetition of results. Besides, we have added some suggestions for urban planning to enhance the sustainability of the RS system based on our results. Please see Section 5 (named 5. Suggestions for urban planning) of the revised version.

5. Comment In the conclusion, the issues include: general statements without theoretical depth on how entropy advances urban sustainability research, over-simplification of findings without addressing limitations, absence of clear policy implications, and structural redundancy where the conclusion repeats earlier content rather than synthesising contributions.

Response: (1) In the conclusion, we have reorganized and theoretically explained how entropy advances urban sustainability research. Please see the Section 6 (named 5. Conclusions) of the revised version.

(2) The limitations of our findings have been added. Please see the Section 6 (named 7. Limitations and future research) of the revised version.

(3) Based the results of our research, we have made policy implications some advice about the sustainability of recreational sports for three cities by taking the suggestion of the expert. Please see the Section 5 (named 5 Suggestions for urban planning) of the revised version.

Looking forward to hearing from you soon.

Yours sincerely,

Xuefang Zou

---

## [Decision Letter · Decision Letter 1]

30 Dec 2025

The sustainability of recreational sports in Chinese cities based on cognitive entropy

PONE-D-25-20453R1

Dear Dr. Chen,

We’re pleased to inform you that your manuscript has been judged scientifically suitable for publication and will be formally accepted for publication once it meets all outstanding technical requirements.

Kind regards,

Genyu Xu, Ph.D.

Academic Editor

PLOS One

Additional Editor Comments (optional):

Reviewers' comments:

Reviewer's Responses to Questions

**Comments to the Author**

Reviewer #2: All comments have been addressed

2. Is the manuscript technically sound, and do the data support the conclusions?

Reviewer #2: Yes

3. Has the statistical analysis been performed appropriately and rigorously?

Reviewer #2: Yes

4. Have the authors made all data underlying the findings in their manuscript fully available?

Reviewer #2: Yes

5. Is the manuscript presented in an intelligible fashion and written in standard English?

Reviewer #2: Yes

Reviewer #2: The manuscript has been sufficiently improved based on the given comments. It has been developed theoretically. The methodological part of the article has also been developed. It has now clearly stated contribution in the article. I can see that the internal validity of the revised manuscript has also been increased. From my point of view, the article is ready for publication.

**Do you want your identity to be public for this peer review?** For information about this choice, including consent withdrawal, please see our Privacy Policy

Reviewer #2: **Yes: ** ROKHSANEH RAHBARIANYAZD

---

## [Editor Report · Acceptance letter]

PONE-D-25-20453R1

PLOS One

Dear Dr. Chen,

I'm pleased to inform you that your manuscript has been deemed suitable for publication in PLOS One. Congratulations! Your manuscript is now being handed over to our production team.

Kind regards,

on behalf of

Dr. Genyu Xu

Academic Editor

PLOS One